# Confinement by COVID-19 and Degree of Mental Health of a Sample of Students of Health Sciences

**DOI:** 10.3390/healthcare9121756

**Published:** 2021-12-19

**Authors:** Arantxa Rymer-Diez, Elisabet Roca-Millan, Albert Estrugo-Devesa, Beatriz González-Navarro, José López-López

**Affiliations:** 1Faculty of Medicine and Health Sciencies, University of Barcelona, 08907 Barcelona, Spain; arantxarymer@gmail.com; 2Oral Health and Masticatory System Group-IDIBELL, Faculty of Medicine and Health Sciences, Odontological Hospital University of Barcelona, University of Barcelona, 08907 Barcelona, Spain; erocamil@gmail.com (E.R.-M.); albertestrugodevesa@gmail.com (A.E.-D.); beatrizgonzaleznavarro@gmail.com (B.G.-N.)

**Keywords:** coronavirus disease, COVID-19, mental health, university students

## Abstract

Background: In response to the global COVID-19 pandemic, most countries have taken important measures to control the spread of the virus, such as population confinement and the closure of universities. Objective: The main objective of this study is to evaluate how the situation resulting from COVID-19 has affected the mental health of a group of health sciences students. Material and Methods: This is a cross-sectional, descriptive, observational study with a sample of 160 people, conducted through an online survey of final-year students of dentistry, nursing and medicine at the Faculty of Medicine and Health Sciences of the University of Barcelona. Results: A total of 82.4% of students reported having suffered stress, anxiety and distress due to the pandemic situation, and 83.10% of participants are worried about not knowing when the academic year will restart. Conclusions: Confinement has negatively affected the mental health of students.

## 1. Introduction

In December 2019, in Wuhan, China, an infectious disease of unknown etiology characterized by severe pneumonia was recognized by the name of “coronavirus disease 2019” (COVID-19) [1]. The microorganism causing this disease has been identified as a new RNA virus of the beta-coronavirus family (SARS-CoV-2) [2,3].

The respiratory disease caused by COVID-19 is highly contagious, and since it spread rapidly, first in China and soon after around the world [4,5,6], the World Health Organization (WHO) therefore decreed that we are facing a pandemic infection [1,7]. Specifically, as of 5 May 2020, a total of 3,517,345 cases and 243,402 deaths had been reported by the WHO [5]. In Spain, on 14 March 2020, a state of alarm and home confinement was decreed since the country registered more than 4200 positive cases and 120 deaths. Therefore, not only has the epidemic brought with it the risks inherent in the disease itself but also a hitherto unknown psychological strain on the population [8,9,10]. Likewise, significant psychological symptoms related to confinement due to COVID-19 have already been identified in the population, such as anxiety, stress, depression, mood swings, irritability, insomnia and attention deficit [4,11].

Although the elders are at greater risk of developing complications from the disease, the study by Naser et al. [1] showed that people of 50 years of age or more had significantly lower risk of developing anxiety compared to the younger population. At the same time, it has been reported that during the pandemic, people with higher levels of education suffer more anxiety, depression and stress. This may be due to the fact that people who have achieved a higher degree of education are likely to be more self-aware in relation to their own health [11].

In response to the global pandemic of COVID-19, universities around the world have ended the academic year on campus by cancelling all on-site events, such as workshops, conferences and sports [2,3], ending the academic year on a non-attendance basis. In Spain, for example, between 9 and 13 March 2020, a progressive closure of schools and universities took place [12].

Distance learning has certain limitations, such as the fact that students can no longer perform clinical procedures and continue their learning by doing [12]. Taking this into account, the pandemic can have a serious impact on university students in terms of teaching and mental health, especially since university students generally have a high incidence of emotional disturbance during the course of their academic life. In addition, it is worth noting the concern that senior students may experience in relation to their incorporation into the world of work, as the uncertainty surrounding the pandemic is increasing [4], especially those with a significant practicum component.

It is also important to emphasize that due to the drastic increase in patients infected by COVID-19 around the world, many medical and nursing students are entering directly into the professional world being part of the group of healthcare professionals destined to deal with the pandemic. This lack of experience in such urgent situations can be stressful for those who graduate in the midst of this exceptional environment [13]. As for dental students, they may also be concerned about the nature of their job, because the main route of transmission of the virus is through micro-droplets and aerosols. It is clear, then, that dentists are among the medical professionals with higher risks of becoming infected and spreading the virus [14].

Another fact to consider is that many young people can be asymptomatic carriers of the disease, and this implies higher concerns among students for putting their relatives at risk, especially the older ones [15].

On the one hand, in the study by Li et al. [13], conducted in health science students, 26.63% of them reported clinically significant distress, while 11.00% met the criteria for acute stress in response to the COVID-19 outbreak. Furthermore, 9.08% showed signs of anguish and acute stress at the same time. On the other hand, in the study by Husky et al. [16], conducted on French university students, the majority (60.2%) of the sample indicated that their anxiety level had increased since the beginning of the confinement period. However, in the study by Cao et al. [8], conducted on medical students at Chagzhi Medical College, the mental health of university students was affected to varying degrees during the outbreak. A total of 75.10% of the students had no symptoms of anxiety, whereas 21.30% had mild anxiety, 2.70% had moderate anxiety and 0.9% had severe anxiety. Therefore, it revealed that 24.90% of university students were affected by anxiety due to the coronavirus outbreak.

In addition to all these factors, health science students have a deeper understanding of the disease and its symptoms, risks and impact on society. Therefore, they could be more susceptible to associated mental disorders during the quarantine period [17].

In view of all the above and given the special situation that the population is facing, the hypothesis raised is that the pandemic situation derived from COVID-19 has affected the mental health of health science students. Therefore, the objective of this article is to evaluate how COVID-19 has affected the mental health of a group of health sciences students at the University of Barcelona. As secondary objectives, this study: (i) evaluates the consumption of psychotropic drugs and other substances in a group of health science students due to various situations triggered by COVID-19 and (ii) evaluates the physical activity of health sciences students during confinement.

## 2. Materials and Methods

### 2.1. Type of Clinical Study

This is a cross-sectional, descriptive, observational study carried out by means of an online survey for dentistry, nursing and medicine students at the Faculty of Medicine and Health Sciences of the University of Barcelona (Bellvitge Campus).

### 2.2. Period of Inclusion and Selection of Participants

The participants of this study were selected during the months of April and May 2020. The final year students of dentistry, medicine and nursing from at the Faculty of Medicine and Health Sciences (Bellvitge Campus) who wished to participate in the study were selected. The inclusion criteria were: (i) Being a student in its a final year of dentistry, medicine or nursing at the Faculty of Medicine and Health Sciences of the University of Barcelona (Bellvitge Campus) and (ii) accepting to participate in the study.

### 2.3. Sample Size

A random sample of 122 individuals is sufficient to estimate, with 95% confidence and a precision of ±5 percentage units, a population percentage that is expected to be around 50%.

### 2.4. Data Collection

Given that the main interest of the study is to evaluate how COVID-19 has affected the mental health of students, an online survey model has been proposed that allows us to evaluate part of the said aspect. This model has been designed through the “Google Forms” portal on 3 April 2020.

In order to carry out the online surveys, the students’ emails were collected manually through the virtual campus, and an email was sent to ask them if they allowed the survey to be sent, and if so, it was sent. All the responses were collected in a file and the data relative to the sending IP were destroyed.

First, the survey informs the participants about the objectives and about the confidentiality of the study, in accordance the Organic Law 3/2018, on the protection of personal data and guarantee of digital rights. The main researcher’s email is also recorded, in case the participants have any questions or want to communicate with the researcher. Moreover, the survey refers to voluntary participation in the study, it consists of a mandatory question in which the survey participants must indicate whether or not they want to participate in the study. If the answer is yes, the participant is sent a 12-question questionnaire, in which it is specified that these questions refer to the situations derived from the COVID-19 pandemic (Table 1).

Finally, we comment that the students did not receive any financial compensation for participating in the study.

### 2.5. Description of the Study Variables

Dichotomous qualitative variables:

–Consumption of psychotropic drugs: Answer Yes/No;

–Sleep disorder: Answer Yes/No;

–Gender: Answer Male/Fame;

–Physical exercise: Answer Yes/No.

Qualitative polycotomic variables:

–Consumption psychotropic drugs: Type response of psychotropic drugs;

–College career: Medicine/Dentistry/Nursing; 

–Students’ sensations: Stress/Anxiety/Anguish/Sadness/Others.

Quantitative variables:

–Age;

–Hours of sleep;

–Hours of exercise.

### 2.6. Statistical Analysis

The data obtained were entered into a Microsoft Excel sheet. Then, they were processed in the SPSS statistical software (version 26, IBM, Armonk, NY, USA) to perform the interferential analysis and the descriptive statistical analysis of the variables studied. The results of the analysis were expressed by bar diagrams (charts) and frequency tables. The Pearson X^2^ correlation coefficient was also used with a significance level of *p*-value ≤ 0.05 and confidence interval of 95% to evaluate the association between the qualitative or categorical variables of the study: (i) gender and affectation by the situations raised, (ii) university degree and impact on the situations raised, (iii) gender and substances use, (iv) university degree and substances use and (v) university degree and type of substances consumed.

## 3. Results

The initial population consisted of 90 dental students, 80 medical students and 90 nursing students, a total of 260 final year students from the Bellvitge University campus. Among these 260 students, 205 students decided to participate, and finally, 160 responses were obtained, representing 61.54% participation.

Among these 160 participants, a total of 80 (50%) are dental students, 41 (25.60%) are nursing students and 39 (24.40%) are medical students. Of the dental students, 82.50% are women and 17.50% are men, with an age average of 24.97 years; of those in nursing, 87.80% are women and 12.20% are men, with an age average of 22.78 years; and of those in medicine, 74.40% are women and 25.60% are men, with an age average of 24.61 years. Therefore, 81.90% women and 18.10% men participated in the study, with an age average of 24.32 years.

With the aims of improving the structure of the results, we have grouped them into five situations derived from the questionnaire: Situation 1: Current situation derived from COVID-19; Situation 2: Not knowing when normality will return again; Situation 3: The impossibility of going out on the streets freely and not seeing their relatives; Situation 4: Do the possible consequences of COVID-19 for your university and/or professional career cause you stress or anxiety?; and Situation 5: Sleep disorder.

### 3.1. Degree of Self-Assessed Impact of COVID-19 on Students

When asked if the current situation derived from COVID-19 (situation 1) had caused them stress, anxiety or other aspects, 16.90% of the participants said no. Among the other 82.30%, “the most repeated words” were stress, anguish and anxiety. They also reported that they felt fear, worry, boredom, uncertainty, sadness and helplessness, among others (Table 2, situation 1 and Table A1). To the question “not knowing when normality (of the academic year) will resume” (situation 2), 83.10% of the participants answered that it caused them anguish, stress or anxiety, and some added words such as anger, depression, uncertainty, despair and worry (Table 2, situation 2; Table A1).

The impossibility of going outside and not seeing family and friends has affected 80.10% of the students (situation 3), producing stress or anxiety in the majority. These also transmitted words such as sadness, anguish and helplessness (Table 2, situation 3; Table A1).

Of those surveyed, 72.50% are concerned about the possible consequences of COVID-19 in their university and/or professional careers (situation 4), feeling stress or anxiety (Table 2, situation 4; Table A1), and 60.00% of the participants admitted (situation 5; Table A1) having sleep disorders during confinement. 

The X^2^ statistical test showed a statistically significant relationship between being a medical student and the affectation of the current situation resulting from the pandemic (*p*-value = 0.02; *p*-value < 0.05). In contrast, no statistically significant relationship was found between university degree and the affectation by the other situations (*p*-value > 0.05; Scenario 2: *p*-value = 0.61, Scenario 3: *p*-value = 0.83; Scenario 4: *p*-value = 0.19; Scenario 5: *p*-value = 0.69; Table A2).

### 3.2. Degree of Self-Assessed Impact of COVID-19 on Students, According to the Gender of the Participants

The current situation resulting from COVID 19 has affected 85.50% of women and 87.00% of men (Table 3, situation 1, and Table A3). Not knowing when the normality (of the academic year) will resume has affected 86.00% of the women surveyed and 72.40% of the men (Table 3, situation 2). The impossibility of going out on the street freely and not seeing their family and friends has affected 81.90% of women and 72.40% of men (Table 3, situation 3). The possible consequences of COVID-19 causes stress or anxiety in 77.90% of women and 48.30% of men (Table 3, situation 4). A total of 64.90% of women and 37.90% of men suffer from sleep disorders (situation 5) during confinement (Table 3, situation 5).

When performing the statistical test X^2^, a statistically significant relationship was obtained between being a woman and being affected by situation 4 and situation 5 (*p*-value < 0.05; Situation 4: *p*-value = 0.00; Situation 5: *p*-value = 0.02). On the other hand, no statistically significant relationship was found between gender and being affected by the other situations raised (*p*-value > 0.05; Situation 1: *p*-value = 0.20; Situation 2: *p*-value = 0.18; Situation 3: *p*-value = 0.67; Table A4).

### 3.3. Substances Use Due to the Situations Raised

A total of 4.40% of health science students consume substances due to the current situation derived from COVID-19 (Table 4, situation 1 and Table A5): 2.50% of the students consumed some substances because they did not know when normality (of the academic year) will resume (Table 4, situation 2); 1.30% of the students used substances due to the impossibility of going out on the streets freely and not seeing their family and friends (Table 4, situation 3); 1.30% of the students consumed some substances due to the possible consequences of COVID-19 for their university and/or professional career (Table 4, situation 4); and 5.60% took substances for sleep disorders (3.00%) (Table 4, situation 5).

When performing the X^2^ statistical test, a statistically significant association was found between being affected by all the situations raised and taking some substances or medication to deal with them (*p*-value = 0.00; *p*-value < 0.05). On the other hand, a statistically significant association was not obtained between the university degree being studied and taking any substances or medication due to the situations raised (*p*-value > 0.05; Situation 1: *p*-value = 0.57; Situation 2: *p*- value = 0.99; Situation 3: *p*-value = 0.85; Situation 4: *p*-value = 0.56; Situation 5: *p*-value = 0.69; Table A6).

### 3.4. Substances Use, by Gender and by the Situations Raised

A total of 4.60% of women consume some substances due to the current situation derived from COVID-19, while no men (0.00%) consume any substances due to this situation (Table 5, situation 1 and Table A7). Moreover, 3.10% of the women consumed some substances because they did not know when academic normality would restart, whereas no men (0.00%) consumed any substances in response to this situation (Table 5, situation 2). A total of 1.50% of the women consumed some substances due to the impossibility of going out on the streets freely and not seeing their family and friends, whilst no man (0.00%) consumed any substances for this reason (Table 5, situation 3). A total of 1.5% of the women consumed some substances due to the possible consequences of COVID-19, while no men (0.00%) consumed any substances for this reason (Table 5, situation 4). Finally, 6.40% of the women and 3.40% of the men asked consumed some substances due to sleep disorders during confinement (Table 5, situation 5).

When performing the X^2^ statistical test, no statistically significant relationship was found between being a woman and taking any medication or substances due to the situations raised (*p*-value > 0.05; Situation 1: *p*-value = 0.45; Situation 2: *p*-value = 0.48; Situation 3: *p*-value = 0.28; Situation 4: *p*-value = 0.28; Situation 5: *p*-value = 0.85; Table A8).

### 3.5. Types of Substances That Students Consume Due to Situations Derived from COVID-19

The types of substances that students consume because of the current situation derived from COVID-19 are diazapam (50.00%), fluoxetine (16.67%) and lormetazepam together with sertraline (16.67%) and valerian (16.67%) (Figure 1, situation 1). By differentiating between university degrees, we obtain the following (Table 6, situation 1): (a) The types of substances that dental students consume for situation 1 are diazepam (50.00%) and fluoxetine (50.00%), (b) the types of substances that nursing students consume due to situation 1 are diazepam (50.00%) and lormetazepam together with sertraline (50.00%) and (c) the type of substance that medical students consume for situation 1 is valerian (100.00%).

The types of substances that students consume because they do not know when academic normality will resume are valerian (33.30%), clonazepam (33.30%) and diazepam (33.30%) (Figure 1, situation 2). When making a division between the university degree, we obtain the following (Table 6, situation 2): (a) The type of substance that dental students consume for situation 2 is diazepam (100.00%), (b) the type of substance that nursing students consume in situation 2 is clonazepam (100.00%) and (c) the type of substance that medical students consume for situation 2 is valerian (100.00%).

The types of substances that students consume due to the impossibility of going outside freely and not seeing their family and friends are clonazepam together with sertraline (50.00%) and diazepam (50.00%) (Figure 1, situation 3). When making a division between the university degree, we obtain the following (Table 6, situation 3): (a) The type of substance consumed by dental students in situation 3 is diazepam (100.00%), (b) the types of substances that nursing students consume in situation 3 are clonazepam together with sertraline (100.00%) and (c) none of the medical students (0.00%) consume any substances due to situation 3.

The types of substances that students consume due to the possible consequences of COVID-19 (e.g., stopping the clinical practicum) for their university and/or professional careers are diazepam (50.00%) and clonazepam together with sertraline (50.00%) (Figure 1, situation 4).

When making a division between the university degree, we obtain the following (Table 6, situation 4): (a) The type of substance that dental students consume in situation 4 is diazepam (100.00%), (b) the types of substances that nursing students consume due to situation 4 are clonazepam together with sertraline (100.00%) and (c) none of the medical students (0.00%) consume any substances due to situation 4. The types of substances that students use for sleep disorders are clonazepam (22.22%), lormetazepam (11.11%), lorazepam (11.11%), melatonin (22.20%) and diazepam (33.30%) (Figure 1, situation 5). When making a division between the university degree, we obtain the following (Table 6, situation 5): (a) The type of substance consumed by dental students in situation 5 is melatonin (100.00%), (b) the types of substances that nursing students consume due to situation 5 are lorazepam (33.33%), melatonin (33.33%) and lormetazepam (33.33%) and (c) the type of substance that medical students consume for situation 5 is clonazepam (50.00%).

### 3.6. Physical Exercise

A total of 81.90% of the students exercise during confinement with an average of 5.09 h per week. When making a division between university careers, we obtain the following (Table 7 and Table A9):

A total of 78.00% of nursing students exercise with an average of 5.21 h per week.

A total of 76.90% of medical students exercise with an average of 5.77 h per week.

A total of 86.30% of dentistry students exercise with an average of 4.75 h per week.

In regard to gender, 87.00% of the female participants and 58.60% of the men exercise. When performing the X^2^ statistical test, a statistically relationship was obtained between being a woman and exercising during confinement (*p*-value < 0.05; *p*-value = 0.00).

## 4. Discussion

### 4.1. Mental Health of Students during Confinement

The current situation resulting from the pandemic has affected the mental health of most of the students surveyed, and their biggest concern is not knowing when the academic year will resume.

The “sensations” that students reported feeling the most in relation to the confinement situation due to COVID-19 were stress, anxiety and anguish, medical students being the group affected most. These data are in line with other publications stating that medical students have a higher prevalence of mental health disorders compared to students from other disciplines [18,19,20]. In any case, it should be noted that the impossibility of going out to the street and seeing family and friends has affected future nurses to a greater degree.

The data obtained in this study is similar to those obtained in various studies conducted throughout surveys, of university students from different countries during confinement due to COVID-19 [4,8,13,15,17].

It is interesting to mention the study by Odriozola et al. [4], carried out in Valladolid (Spain) two weeks after confinement due to the COVID-19 pandemic, in which 34.19% of the participants reported symptoms of depression, 21.34% reported anxiety symptoms and 28.14% reported stress symptoms [4]. In this study, significantly higher depression, anxiety and stress scores were reported in college students compared to different groups of college workers [4]. Moreover, the study revealed that, as opposed to what was found in other different studies [1,15], confinement has affected more to students of artistic sciences, humanities and social sciences than to those of health sciences and engineering and architecture.^3^ In a similar study, the one of Liu et al. [17], conducted on medical students of “Tongji Medical College” Wuhan (China)**,** 35.50% of the survey participants were in a state of depression and 22.10% were in a state of anxiety. Most of them suffered from mild or moderate states of the aforementioned pathologies.

The studies of Odriozola et al. [4] and that of Liu et al. [17] found that the percentage of students with symptoms of depression was higher than the percentage of students with symptoms of anxiety, stress or anguish. Nonetheless, in the current study, it was found that a higher percentage of university students had symptoms of anxiety, stress or anguish.

### 4.2. Substances That the Students Consumed during Confinement 

The substances that students reported taking the most due to situations derived from COVID-19 were psychotropic drugs, specifically anxiolytics and antidepressants. They also reported taking melatonin for sleep disorders and valerian, but to a lesser extent.

In a study carried out in health sciences students on the usual consumption of psychotropic drugs by university students, 16.40% of the students used psychotropic drugs [21]. Likewise, in another study carried out between 2012 and 2013 at the University of Dusselford [19] only in medical students, 10.70% of these reported that they used psychotropic drugs. Therefore, if we compare the data from these two studies with the results of the present study, an increase in the consumption of psychotropic drugs due to confinement is not observed.

Unlike two studies [21,22] in which antidepressants were found to be the most consumed psychotropic drugs among university students, in the current study it was observed that anxiolytics were the most consumed by students to face this situation, specifically diazepam. However, its consumption due to this situation is lower than the usual consumption of anxiolytics in university students, since, in the study by Martínez et al. [21], 5.00% of students consumed them, and according to Romero et al. [23], the consumption of benzodiazepines in medical students from the fifth year of university can increase to 32.40% in the seventh year.

### 4.3. Physical Activity 

Students in the healthcare setting are educated about the health benefits of physical activity, but this knowledge does not always translate into their personal lifestyle choices. In various studies, it has been reported that health science students perform low levels of physical activity [24,25]. For example, in the study by Blake et al. [24] conducted in nursing and medical students in the United Kingdom, 48.00% of medical students and 38% of nursing students did not achieve the 150 min of moderate intensity physical activity recommended per week. The high levels of inactivity found are similar to the results revealed by previous studies that used similar samples [25]. This shows, regardless of the current situation, a clear need to promote physical activity among these students.

In spite of that, in the present study, it can be observed that 81.90% of the students carried out physical exercise during confinement with an average of 5.09 h per week. Thus, they exceeded the 150 min of recommended physical activity. These data contradict a similar study carried out in the United States on university students during confinement, in which they found that the students decreased their physical activity during confinement [6].

The increase in physical activity among the students of this work could be related to an attempt to reduce stress, taking into account the well-known relationship between physical exercise and improved stress levels. In this sense, in the study conducted by Blake et al. [24], it was found that medical and nursing students believe that one of the benefits of exercising is stress relief. Likewise, the reason for physical exercise may be based on the “desire” of the students to break the monotony of confinement or to feel fulfilled by the mere fact of exercising. It is also logical to think that the reason may be related to the free time available since, according to what is reported by the students, the lack of time is one of the factors that prevents them from doing physical exercise.

In this study, it was found that dental students (86.30%) were the ones who exercise more during confinement, followed by nursing students (78.00%) and, lastly, those of medicine (76.90%). Certainly, the fact that medical students are the ones who, on the one hand, have been most affected by the COVID-19 situation and, on the other hand, those who have exercised the least, are striking. However, this relationship was not statistically significant (*p*-value < 0.05).

### 4.4. Gender

A statistically significant relationship was obtained between being a woman and suffering from sleep disorders during confinement in addition to between being a woman and suffering stress or anxiety due to the possible consequences of COVID-19 in the university and/or professional career.

Similarly, in the study by Naser et al. [1], conducted in Jordania on university students during confinement, it was found that women had a higher risk of anxiety and depression, which is in agreement with the data reported during other epidemics in which women, particularly those who work in healthcare, were prone to developing depression and anxiety, since, according to various studies [1,26,27], female gender is a risk factor for developing mental illnesses. In contrast, in the study by Liu et al. [17] and Cao et al. [8], there were no significant gender differences in the prevalence of depression and anxiety, and the two sexes were equally psychologically affected by the COVID-19 epidemic [8,17].

It is interesting to note that in the current study, women exercise more than men during confinement and the difference was statistically significant (*p*-value < 0.05; *p*-value = 0.00).

### 4.5. Sleep Disorder

Not only can the situation of home confinement increase the levels of stress, anxiety and depression, but it can also disrupt sleep. It is important to highlight that due to the fundamental role that sleep plays in regulating emotions, altering it can have important consequences [28].

In this study, 60% of the participants suffered from sleep disturbances due to confinement. This may be due to the inactivity they had during the day [28] and the concern they have reported having for this situation. According to various articles, stress causes many difficulties in sleeping, such as restless sleep, waking up in the middle of the night or getting up too early [28,29]. In this sense, in the present study, it was found that there is a statistically significant association between poor sleep and suffering from stress. Likewise, in Almojali et al.’s study [30] in medical students, a high prevalence of poor sleep quality (76.00%) and stress (53.00%) was found with a statistically significant association (*p* < 0.00). It was also indicated that students who do not experience stress are less likely to have poor quality sleep [29].

Regarding the consumption of substances to sleep, in this study, 5.60% of the students took substances to sleep during confinement. This percentage is slightly higher than the one observed by Wege et al.’s study [19], in which 4.60% of the students stated that they took substances to improve sleep.

It is also interesting to note that the students in the present study consumed more substances to combat sleep disorders during confinement than for the other situations arising from COVID-19.

Finally, it is worth mentioning that we would like this study to be a reason to initiate research on this topic in different university settings and also to serve as a motivation to develop a validated tool specifically created to enquire about “mental health during confinement”, adding, if possible, questions about “post-confinament” so that it is easier to correctly compare the different studies carried out with similar samples in different countries and be able to draw reliable conclusions. Although there are some studies that use the validated GAD-7 survey, the questions they use are different from ours and from other studies on this topic since it is not a specific questionnaire about “mental health during confinement”, and with this validated tool, it is not possible to study everything related to this topic. Lastly, from our point of view, the study is helpful to reflect about an issue as important as mental health among young people.

### 4.6. Limitations

The study has several limitations. A non-validated survey is used because at the time that we made the survey (April 2020) no previous surveys on this topic were found. It is a self-reported study, the sample is small and it was made from a single university setting (health sciences students). It is also important to highlight that this type of study does not allow us to determine causality, but we can simply make a descriptive analysis of the data obtained in the surveys and, through the chi-square, obtain relationships between the variables analyzed and confinement.

## 5. Conclusions

Confinement has affected students negatively, and it could be interesting, in the future, to carry out more studies with larger samples, in different university settings and different universities on the ways in which COVID-19 has affected their mental health. Students have not increased their intake of substances or psychotropic drugs due to situations derived from COVID-19. Most of the students (81.90%) have exercised more than usual during confinement. Due to the abovementioned limitations, the results of the study should be interpreted with caution.

## Figures and Tables

**Figure 1 healthcare-09-01756-f001:**
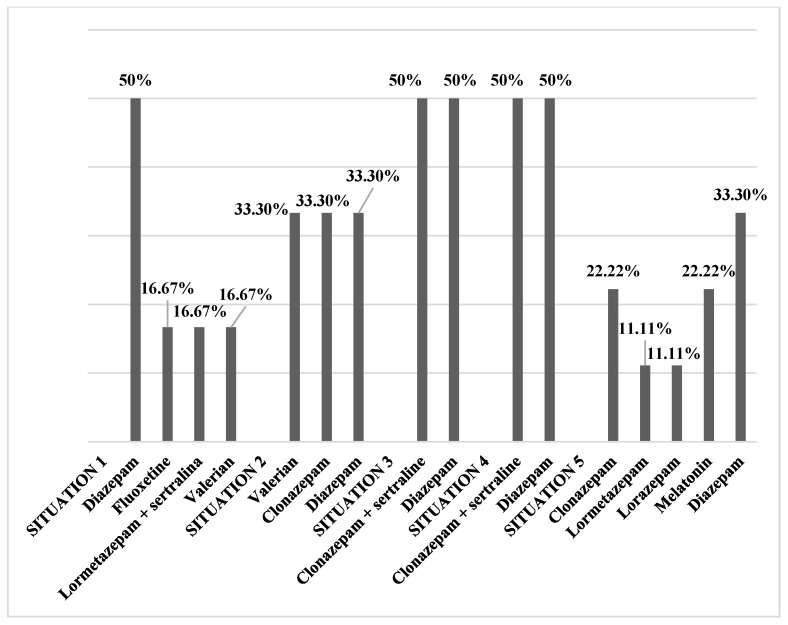
Substances that students have reported taking due to situations derived from COVID-19.

**Table 1 healthcare-09-01756-t001:** Questionnaire given to students about COVID-19.

Age:
Gender: Male/Female
College Degree: Medicine/Nursing/Dentistry
1. The current situation derived from COVID-19 has produced you: Stress/Anxiety/Anguish/Sadness/Others/None
2. Do you take any substances or medicine for the situation described in question 1? Yes/No Which one?
3. Has this situation produced you a sleep disorder? Yes/No
4. Do you take any substances or medicine for the situation described in question 3? Yes/No Which one?
5. Not knowing when academic normality (the academic year) will resume causes you: Anguish/Stress/Anxiety/Sadness/None/Other
6. Do you take any substances or medicine for question 5? Yes/no Which one?
7. Has the impossibility of going out on the streets freely and not seeing your family and friends caused you? Anguish/Stress/Anxiety/Sadness/Others/None
8. Do you take any substances or medication for question 7? Yes/No Which one?
9. Do the possible implications (e.g., no more internships) of COVID-19 for your university and / or professional career cause you stress or anxiety? Yes/No
10. Do you take any substances or medication for question 9? Yes/No Which one?
11. Do you practice any type of physical exercise on these days of confinement? Yes/No
12. Nº hours per week:

**Table 2 healthcare-09-01756-t002:** Sensations, according to university degree, that participants reported feeling due to the different situations that COVID-19 has caused.

Different Situations That COVID-19 Has Caused	University Degree			
The current situation derived from COVID-19 has caused in you (situation 1)	Nursing	Medicine	Dentistry	TOTAL
Affectation by this situation	80.30%	90.40%	80.50%	82.30%
Blank	0.00%	0.00%	1.30%	0.60%
None	19.50%	10.30%	18.80%	16.90%
Not knowing when normality return has caused in you (situation 2)	Nursing	Medicine	Dentistry	TOTAL
Affectation by this situation	75.40%	89.80%	83.00%	83.10%
None	22.00%	10.30%	15.00%	15.60%
blank	2.40%	0.00%	1.30%	1.30%
The impossibility of going out on the street freely and not seeing your family and friends has caused in you (situation 3)	Nursing	Medicine	Denistry	TOTAL
Affectation by this situation	89.90%	85.00%	73.00%	80.10%
None	9.80%	15.40%	26.30%	19.40%
Blank	0.00%	0.00%	1.30%	0.60%
The possible consequences of COVID-19 (e.g., no more internships) of COVID-19 for your university and/or professional career causes you stress or anxiety (Situation 4)	Nursing	Medicine	Dentistry	TOTAL
No	41.50%	17.90%	23.80%	26.90%
Yes	58.50%	82.10%	75.00%	72.50%
Blank	0.00%	0.00%	1.30%	0.60%
Sleep disorder (situation 5)	Nursing	Medicine	Dentistry	TOTAL
No	34.10%	46.20%	38.80%	39.4%
Yes	65.90%	53.80%	60.00%	60.00%
Blank	0.00%	0.00%	1,30%	0.60%

**Table 3 healthcare-09-01756-t003:** Affection of COVID-19 according to gender.

Different Situations That COVID-19 Has Caused	Gender	
The current situation derived from COVID-19 has affected you (situation 1)	Female	Male
Yes	85.50%	87.00%
Not knowing when normality will resume again has affected you (situation 2)	Female	Male
Yes	86.00%	72.40%
The impossibility of going out on the street freely and not seeing your family and friends has affected you (situation 3)	Female	Male
Yes	81.90%	72.40%
The possible consequences of COVID-19 for your university and / or professional career have caused you stress or anxiety (situation 4)	Female	Male
Yes	77.90%	48.30%
Sleep disorder (situation 5)	Female	Male
Yes	64.90%	37.90%

**Table 4 healthcare-09-01756-t004:** Percentage of students who consume substances due to the situations raised, according to university degree.

Take any Medication or Substances	University Degree			
Do you take any substances for the current situation derived from COVID-19 (situation 1)?	Nursing	Medicine	Dentistry	TOTAL
Yes	7.30%	2.60%	2.50%	4.40%
Do you take any substances because you do not know when normality will resume (situation 2)?	Nursing	Medicine	Dentistry	TOTAL
Yes	2.40%	2.60%	2.50%	2.50%
Do you take any substances due to the impossibility of going out on the street freely and not seeing your family and friends (situation 3)?	Nursing	Medicine	Dentistry	TOTAL
Yes	2.40%	0.00%	1.30%	1.30%
Do you take any substances due to the possible consequences of COVID-19 (e.g., no more internships) for your university and/or professional career (Situation 4)?	Nursing	Medicine	Dentistry	TOTAL
Yes	2.40%	2.60%	0.00%	1.30%
Do you take any substances for sleep disorders (situation 5)?	Nursing	Medicine	Dentistry	TOTAL
Yes	7.30%	10.30%	3.00%	5.60%

**Table 5 healthcare-09-01756-t005:** Percentage of students who consume substances due to the situations presented, according to gender.

Take any Medication or Substances	Gender		
Do you take any substances for the current situation derived from COVID-19 (situation 1)?	Female	Male	TOTAL
Yes	4.60%	0.00%	3.80%
Do you take any substances because you do not know when normality will resume (situation 2)?	Female	Male	TOTAL
Yes	3.10%	0.00%	2.50%
Do you take any substances due to the impossibility of going out on the street freely and not seeing your family and friends (situation 3)?	Female	Male	TOTAL
Yes	1.50%	0.00%	1.30%
Do you take any substances due to the possible consequences of COVID-19 (e.g., no more internships) for your university and/or professional career (situation 4)?	Female	Male	TOTAL
Yes	1.50%	0.00%	1.30%
Do you take any substances for sleep disorders (situation 5)?	Female	Male	TOTAL
Yes	6.10%	3.40%	5.60%

**Table 6 healthcare-09-01756-t006:** Substances that students have reported taking according to university degree.

Types of Substances That Students Take for the Situations Raised	Dentistry	Nursing	Medicine
Situation 1	50.00% diazepam	50.00% diazepam	100.00% valerian
	50.00% fluoxetine	50.00% lormetazepam + sertralina	
Situation 2	100.00% diazepam	100.00% clonazepam	100.00% valerian
Situation 3	100.00% diazepam	100.00% clonazepam + sertralina	0.00%
Situation 4	100.00% diazepam	100.00% clonazepam + sertralina	0.00%
Situation 5	100.00% melatonin	33.33% lorazepam	50.00% melatonin
		33.33% melatonin	50.00% clonazepam
		33.33% lormetazepam	

**Table 7 healthcare-09-01756-t007:** Percentage and number weekly hours of physical activity.

Exercise during Confinament	University Degree			
Do You Do Exercise?	Nursing	Medicine	Dentistry	TOTAL
Yes	78.00%	76.90%	86.30%	81.90%
Number of hours per week	Nursing	Medicine	Dentistry	TOTAL
Mean	5.21	5.77	4.75	5,09
Median	5	6	5	5
Mode	3	3	4	6
Minimum	2	1	1	1
Maximum	10	14	14	14

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
