# Peer review of "Confinement by COVID-19 and Degree of Mental Health of a Sample of Students of Health Sciences"

_healthcare, 2021, doi:10.3390/healthcare9121756_

Round 1

Reviewer 1 Report

Here are some comments about the manuscript. Despite an interesting topic and population, there are some methodological weaknesses and some lack of information about procedure. However, the introduction and discussion parts are well documented.

-l21: (a) for "(a)re worried" and not "re worried"

-the method to compute sample size is not clear: on what method did they rely to make this assumption? (A random sample of 155 individuals is sufficient to estimate, with 95% confidence 95 and a precision of +/- 5 percentage units, a population percentage that is expected to be 96
around 50%.) (l94)

-Where did the questions come from (i.e. table 1)?  The choice of tools to assess mental health of students is not based on  scales validated scientifically.

- A lot of parts in discussion should be moved to introduction part to better justified the present study (e.g. l. 296 ti l.306)

Reviewer 2 Report

Thank you for inviting me to review this manuscript, titled “Confinament by Covid-19 and degree of mental health of a group of students of health sciences.”.  This study propose to examine how the situation resulting from COVID-19 has affected the mental health of a group of health sciences students.

The main question addressed by the research is relevant and interesting. The topic is original. The specific issue or problem is defined.

The proposed objective is relevant, as well as the evidence provided and the accompanying discourse. It gathers coherent and argued information at the different key points of the process. The research design, questions, hypotheses, and methods are clearly established.

The characteristics and the inclusion and exclusion criteria of the sample are adequately described. The analyzes used are justified and presented in an understandable way. Most instruments with adequate reliability and validity psychometric properties were administered for this.

However, the manuscript would benefit from some small suggestions or changes. See specific suggestions below.

Regarding the title, remove the period at the end and check if, with a view to greater generalization, it is possible to change the word group of students to students or in a sample of…

It would be necessary to alphabetize the keywords in the abstract

If it is possible to indicate a theoretical model that more precisely integrates the variables of the study

I would add a section expanding the information on the type of design used

I would add a study in which it is indicated that the invalid instrument contains questions that are usually relevant in this type of study.

Some statistics such as p are written in italics within the text, in tables, etc.

I would not use zero before decimal point when the number cannot be greater than 1 as is the case with probabilities

Make reference to the fact that the Study has the approval of the Institution's Ethics Committee, etc.

Expand the section on future lines of research and conclusions

It would further expand the future lines of research and a paragraph with the importance and implications derived from the study as well as the conclusions section

Adjust all references to journal standards, eg year in bold, etc.

Reviewer 3 Report

The authors cross-sectional, descriptive, observational study(online survey)) with final year students.

The subject is relevant, the methodology simple. Data on different populations that might suffer from COVID threat or restriction measures are relevant.

The paper might be improved, by giving a description of the actual situation in the site chosen, i.e. infection /death rates, restrictions implemented in the region and University in more detail , some data on the University etc.

The questionnaire is not combined with any standard questionnaire, and the sample limited to one University, so its difficult to compare or draw general conclusions on students in different Universities or countries.

The focus on medication use gives relevant results, though the questionnaire was based on general considerations, not on prior qualitative research and does not yield data on all possible problems or coping strategies.

A style/grammar check is recommended, though the general style is good- see for example sentence structure

, being medical students 288
the most affected

better: medical students being the group affceted most ?

The statistics looks ok, but a separate statistical review might be advisable.

Round 2

Reviewer 1 Report

Dear authors,

Thank you for your revisions. You answered are satisfying and you provide sufficient information in your manuscript to better understand all of your work.

Kind regards.

Reviewer 2 Report

The manuscript has improved a lot with the suggestions made.